# Detection and Phylogenetic Analysis of a Novel Tick-Borne Virus in Yunnan and Guizhou Provinces, Southwestern China

**DOI:** 10.3390/pathogens10091143

**Published:** 2021-09-05

**Authors:** Anan Wang, Zheng Pang, Lin Liu, Qianwen Ma, Yize Han, Zhijie Guan, Hao Qin, Guoyu Niu

**Affiliations:** 1Key Laboratory of Health Inspection and Quarantine, School of Public Health, Weifang Medical University, Weifang 261053, China; wangananweiyi@163.com (A.W.); maqw2021@163.com (Q.M.); hanyize2000@163.com (Y.H.); gZJ15163605425@163.com (Z.G.); 2Infectious Disease Drug Discovery Institute, Tianjin International Joint Academy of Biomedicine, Tianjin 300457, China; pangzheng@tjab.org; 3Immune-Path Biotechnology (Suzhou) Co., Ltd., Suzhou 215000, China; liulin2016bj@163.com

**Keywords:** Dabieshan tick virus, ticks, *R. microplus*, *H. longicornis*, Yunnan Province, Guizhou Province

## Abstract

Dabieshan tick virus (DTV) is a novel tick-borne virus with the potential to infect both animals and humans. It has been confirmed that DTV is widely distributed in Shandong and Zhejiang Provinces. In this study, a total of 389 ticks were sampled from Honghe city of Yunnan Province and Bijie city of Guizhou Province, and then divided into 148 pools according to the location and species. QRT-PCR and nested PCR were performed to confirm the presence of DTV. The results showed a minimum infection rate of 2.43% (5/206) in Yunnan Province and 3.28% (6/183) in Guizhou Province, respectively. Interestingly, DTV was identified in *Rhipicephalus*
*m**icroplus* for the first time besides *Haemaphysalis longicornis*. Phylogenetic analysis showed that DTV from Yunnan and Guizhou Provinces shared over 94% identity with isolates derived from Hubei and Shandong Provinces, and DTV was relatively conservative in evolutionary dynamics. These findings provide molecular evidence of Dabieshan tick virus in different species of ticks from unrecognized endemic regions and suggest that DTV may be widely prevalent in southwestern China.

## 1. Introduction

Tick-borne viruses (TBVs) are a large group of viruses that can be transmitted by tick bites [1]. Currently, the TBVs known to cause human disease belong mainly to five families, such as Nairoviridae, Phenuiviridae, Flaviviridae, Orthomyxoviridae, and Reoviridae [2]. In recent years, emerging TBVs, including severe fever with thrombocytopenia syndrome virus (SFTSV), Heartland virus (HRTV), Bourbon virus (BRBV), and Alongshan virus (ALSV) have been identified one after another, often with substantial impacts on public health and severe clinical symptoms [3,4,5,6]. These TBVs are transmitted from their natural hosts to human beings through zoonotic infection sources, resulting in disease of varying severity or even death. Thus, it is of great importance to strengthen the identification, investigation, and pathogenicity research of emerging TBVs for better understanding of these viruses and handling of new or re-emerging related diseases.

Dabieshan tick virus (DTV), which belongs to Uukuvirus genus, Phenuiviridae, Bunyavirales, is one of the novel TBVs and was first identified in *H*. *longicornis* from Hubei Province in 2015 [7]. At present, there are few researches on DTV, especially the vector ecology, host range, transmission route, distribution range, etc. However, DTV was reported to be widely distributed in ticks from Shandong Province and Zhoushan area [8,9], and there was no significant difference in nucleotide sequence between those isolates and Hubei Province, indicating that DTV was relatively conservative in evolution. Therefore, it is necessary to further investigate the entomological, virological, and epidemiological features of this virus.

Yunnan and Guizhou Provinces are located in southwestern China, geographically belonging to the Yunnan-Guizhou Plateau and subtropical humid area. These areas are rich in natural resources, diverse vegetation types, and abundant rainfall, which are suitable habitats for tick survival. In the present study, unfed and engorged ticks were sampled at two locations in Yunnan and Guizhou Provinces for the detection of DTV, and the result revealed the occurrence of DTV in southwestern China for the first time. In addition, it was proved that DTV was relatively conservative in evolution by sequence alignment and evolutionary analysis.

## 2. Results

### 2.1. Tick Collection

In total, 116 ticks of the species *H. longicornis* (95, 81.9%) and *R. microplus* (21, 18.1%) were collected from cattle hosts. All ticks collected from animals were engorged adults. Meanwhile, 273 ticks were collected by dragging a cloth through vegetation, including *H. longicornis* (224, 82.1%, 224 adults, 0 nymphs, and 0 larvae) and *R. microplus* (49, 17.9%, 49 adults, 0 nymphs, and 0 larvae). *H. longicornis* was the dominant species. Geographically, 206 ticks in Yunnan Province and 183 ticks in Guizhou Province were sampled (Table 1).

In total, 389 ticks from two species were collected in the study region. The ticks were identified and grouped into 148 pools by site, host, species, and collection type in preparation for the detection of viral RNA.

### 2.2. Detection of DTV RNA in Ticks 

Of the species collected, both *H. longicornis* and *R. microplus* were positive for DTV. Of which, DTV was detected with the minimum infection rate (MIR) of 2.83% (95% CI = 1.18–4.47), based on 11 positive pools out of 148, containing a total of 389 ticks (Table 1). Geographically, the MIR of DTV in ticks from Yunnan and Guizhou Provinces was 2.43% (5/206) (95% CI = 0.33–4.53) and 3.28% (6/183) (95% CI = 0.70–5.86), respectively. No significant difference was found in the prevalence of DTV among ticks collected from the two sampling sites. Unfortunately, the virus was not isolated from any positive tick samples.

### 2.3. Phylogenetic Analysis

In this study, the partial S segment of DTV was successfully amplified and sequenced from 5 of 11 qRT-PCR-positive samples. Consistent with a previous report, phylogenetic analyses of the S segment of these DTV isolates indicated that these viral isolates clustered with Yongjia tick virus 1 and Uukuniemi virus. Furthermore, these partial sequences of S segment were compared with published DTV sequences obtained from *H. longicornis* in Hubei and Shandong Provinces. The results showed that these DTV sequences obtained from Yunnan and Guizhou Provinces were genetically close to the known sequences reported from the above-mentioned locations (Figure 1 and Figure 2). Pairwise distances analysis showed that these DTV sequences obtained in this study had a 97.3–99.1% nucleotide identity with each other and all the sequences from different regions of China shared more than 94% identity, which demonstrated a close evolutionary relationship among those DTV sequences.

## 3. Discussion

In recent years, more and more novel arboviruses have been identified from various vector hosts [10,11]. Additionally, a growing number of novel TBVs were also discovered throughout the world, for instance, BRBV in USA [4], ALSV in China [5], Hunter island group virus (HIGV) in Australia [12], Muko virus (MUV), and Tarumizu tick virus (TarTV) in Japan [13,14], etc. Therefore, an extensive investigation of viruses in ticks is propitious to penetrate into the diversity of TBVs and deal with new or re-emerging Tick-borne virus diseases. Yunnan-Guizhou Plateau is one of the richest areas of global biodiversity and ecological landscape, which is located in a remote area with low human population density and extremely abundant insect fauna and species distribution, supporting circulation of numerous arthropod-borne viruses [15]. 

In this study, DTV was first identified in Yunnan and Guizhou Provinces. Epidemiological investigations showed that DTV may be widely distributed in Yunnan-Guizhou Plateau. However, there was no significant difference of DTV prevalence in ticks between these two sites. By comparison with the previous survey about DTV in ticks from Shandong Province [8], the MIR was relatively high (2.83%) in this study. This may be due to the high carrying rate of DTV in ticks from a local area or sampling deviation. In addition, our result indicated a high prevalence of DTV in engorged ticks from cattle (5.17%, 6/116), compared with that of unfed ticks from vegetation (1.83%, 5/273) in the same region. It seemed probable that domestic animals could enhance virus replication or play a positive role in virus spread.

Notably, DTV was first identified in *R. microplus* in this study, which is also the first time that DTV was found in other tick species except *H. longicornis.* Both tick species could bite humans, so there is a risk of transmitting DTV to humans. Our results extended the tick-borne infection spectrum of DTV and provided a viewpoint for further understanding the natural circulation characteristics of this virus. Owing to the insufficiency in clinical research about this novel virus, the pathogenicity of DTV is still unknown. 

A phylogenetic tree was constructed by the neighbor-joining method based on the partial nucleotide sequences of S segment obtained in this study and other reference sequences from GenBank. Phylogenetic analysis showed that these five strains of DTV in this study were closely related to the isolates found in Hubei and Shandong Provinces [7,8], indicating that the sequences of DTV from different regions had high similarity. These results demonstrated that DTV had a relatively low mutation rate and was relatively conservative in evolution. However, more sequence information of other fragments, such as L segment and M segment, should be further verified. According to our estimate, as an ancient natural focus virus, DTV is widely distributed through China and transmitted by tick biting. Nevertheless, the pathogenicity of DTV and the role of animals and humans in the virus life cycle remain to be further illustrated.

Additionally, the limitation of our research should be noted. Firstly, no viable virus was successfully isolated in this study. However, this result cannot be construed as an evidence of absence of the virus, and it may due to the extremely low viral titers in these ticks. Secondly, there was a lack of the investigation on DTV infection in local animals.

## 4. Methods

### 4.1. Ticks Collection and RNA Extraction 

From July through August 2019, ticks were collected from both animals and vegetation in the Honghe region, Yunnan Province (102°54′ E, 24°02′ N) and Bijie region, Guizhou Province (105°49′ E, 27°22′ N) (Figure 3). Cattle were the most common domesticated animal species in these regions and adult cattle raised in free range were selected in this study. The attached ticks from cattle were collected using forceps and placed in perforated tubes containing a moistened piece of filter paper. All the engorged ticks collected from animals were stored in a cool and ventilated place for 1 week, and then transferred into liquid nitrogen. Meanwhile, ticks from vegetation were collected by “the woolen flannel cloth dragging method” as described by Mejlon and Jaenson. All the ticks collected from vegetation were unfed and stored in liquid nitrogen. Preliminarily, we determined the species of ticks according to morphology, and the result was verified by the detection of the tick conserved gene: *COI*. The PCR primer sets used in this study are LCO1490 (GGT CAA CAA ATC ATA AAG ATA TTG G) and HCO2198 (TAA ACT TCA GGG TGA CCA AAA AAT CA). All the unfed ticks were grouped into pools of 5–15 individuals each by collection site, species, and stage while engorged ticks were processed individually. 

Individual or pooled tick specimens were surface-sterilized with sequential washes with Dulbecco’s modified eagle’s medium (DMEM) containing antibiotics and then homogenized in 500 μL chilled DMEM using a tissue homogenizer (Qiagen, Hilden, Germany). The tick homogenates were transferred into 1.5 mL microcentrifuge tubes and centrifuged at 4 °C and 10,000× *g* for 5 min (Eppendorf, Hamburg, Germany). Then, the clarified supernatants of tick homogenates were used for RNA extraction using the TIANamp RNA extraction kit (Tiangen, Beijing, China) following the manufacturer’s instructions.

### 4.2. PCR for Detection of Pathogen in Ticks 

QRT-PCR and nested PCR were respectively performed to confirm the presence of DTV and gain the sequence information. The primer sets and a probe targeting the S segment of DTV used in this study are DTV-F (TGC TCC TCT CCG CAC ACC T), DTV-R (TGG CAA GTA GAG GAA ACT GGT GA), and DTV-P (FAM-TCC CTC CAG CCA TCA CCA CCT CC-BHQ1). The nested PCR primer sets targeting the S segment of DTV used in this study are Out-F (GGC AGC ACT TTC ACG GAT G), Out-R (CCC CTG TCA TGT CTA ATC AAT GG), In-F (GCA AGC AGA GCC TCA AGA AGC), and In-R (GCC AGA TTG CGA TCC AAG TAT G). The amplified products were visualized by SYBR^®^ Safe (Thermo Fisher Scientific, Waltham, MA, USA) after 1% agarose gel electrophoresis. 

### 4.3. Virus Isolation

Tick homogenates which were positive for DTV viral RNA by qRT-PCR were subjected to centrifugation, and the supernatant was passed through a sterile 0.45 μm filter (Merck Millipore, Billerica, MA, USA). Then, the filtrates were subjected to virus isolation using the cell line BHK–21 cultured in six-well plates. Briefly, 100 μL of sample was inoculated onto a monolayer of BHK–21 cells at 37 °C in 5% CO_2_ conditions for 7 days. Cells were inspected daily and tested for the presence of DTV RNA by qRT–PCR on day 6–8 post-infection. After three additional blind passages, the supernatants were harvested and cryopreserved at -80 °C until further analysis. Virus isolation was performed at a BSL–2 laboratory in School of Public Health, WeiFang Medical University.

### 4.4. Data Analyses

The prevalence of DTV in ticks was presented as the minimum infection rate (MIR) with a 95% confidence interval (95% CI). Fisher’s exact test was performed to evaluate the statistical difference in the positive rate. Differences with *p* < 0.05 were considered to be statistically significant. SeqMan and MegAlign of the Lasergene software package (DNASTAR, Madison, WI, USA) were used to edit and align the obtained sequences. The MEGA5.1 software was utilized for phylogenetic analysis by constructing the neighbor-joining evolutionary tree with 1000 replicates, compared with previously published viral sequences of DTV strains and other *Uukuvirus genus* strains, such as Rukutama virus, Kabuto mountain virus, Uukuniemi virus, Silverwater virus, etc. The relevant GenBank accession numbers have been marked.

## 5. Conclusions

In conclusion, our data provided important evidences for the occurrence of DTV in ticks in a non-endemic area, in southwestern China. Specially, this virus was identified in *R.*
*m**icroplus* for the first time, suggesting that DTV may be transmitted through more tick species. In short, further studies are required to elucidate the pathogenicity and epidemiological characteristics of DTV.

## Figures and Tables

**Figure 1 pathogens-10-01143-f001:**
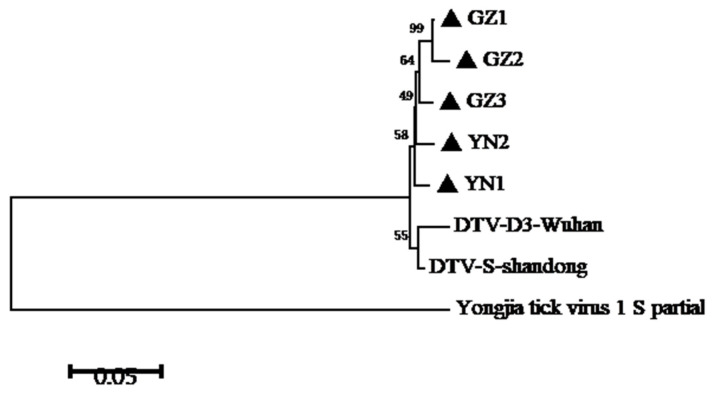
Phylogenetic analysis of DTV S-segment sequences amplified from ticks of Yunnan and Guizhou Provinces. A phylogenetic tree based on S-segment sequences (548bp) by the NJ method using MEGA 5.1 is shown. DTV-D3-Wuhan indicates the tick-derived sequence amplified from Hubei Province in 2015. DTV-S-Shandong indicates the tick-derived sequence amplified from Shandong Province in 2018.

**Figure 2 pathogens-10-01143-f002:**
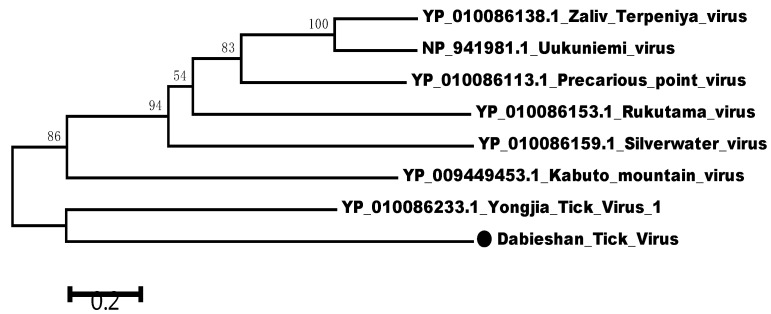
Phylogenetic relationships between DTV and other Uukuvirus genus viruses based on the amino acid sequence similarities of viral S segment. Phylogenetic analysis was performed based on the partial sequence of viral S segment (NS gene) using the neighbor-joining method. The percentages of 1000 bootstrap replication are indicated at the nodes. The DTV is indicated with a black dot.

**Figure 3 pathogens-10-01143-f003:**
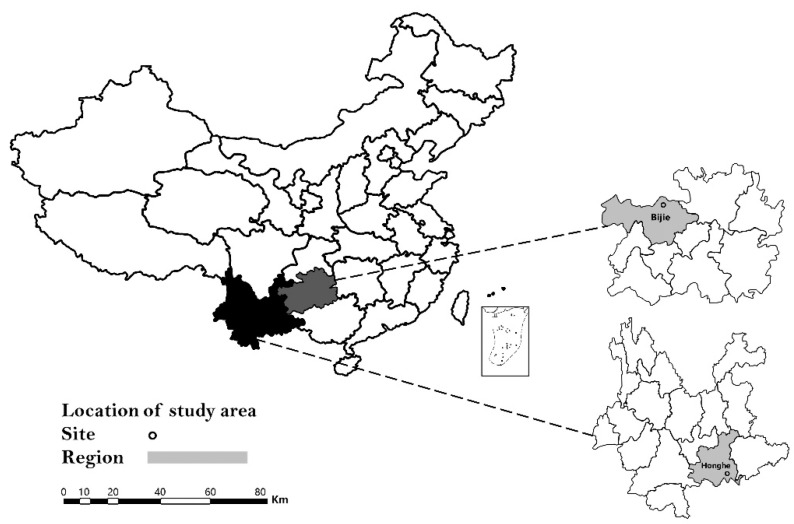
Location of Yunnan and Guizhou Provinces in China (left), and location of Honghe and Bijie city within the respective province where tick samples were collected in 2019.

**Table 1 pathogens-10-01143-t001:** Detection of DTV viral RNA in ticks collected from Yunnan and Guizhou Provinces by origin and species.

		Yunnan Province				Guizhou Province			
	Origin	No. of Ticks	Pools	qRT-PCR	MIR%	No. of Ticks	Pools	qRT-PCR	MIR%
*H. longicornis*	Cattle	52	52	2	3.84	43	43	3	6.98
	Dragging	84	10	1	1.19	140	17	3	2.14
Subtotal		136	62	3	2.21	183	60	6	3.28
*R. microplus*	Cattle	21	21	1	4.76	/	/	/	/
	Dragging	49	5	1	2.04	/	/	/	/
Subtotal		70	26	2	2.86				
Total		206	88	5	2.43	183	60	6	3.28

“/” indicates that no ticks were collected.

## Data Availability

All data generated or analyzed during this study are included in this published article. Access to raw data can be acquired by contacting the corresponding author via email.

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
