# Peer review of "Detection and Phylogenetic Analysis of a Novel Tick-Borne Virus in Yunnan and Guizhou Provinces, Southwestern China"

_pathogens, 2021, doi:10.3390/pathogens10091143_

Round 1

Reviewer 1 Report

The manuscript of Wang et al. reports molecular detection of Debiashan tick virus (DTV) in ticks sampled from vegetation and cattle in two locations in southwestern China. DTV was detected in questing and partially engorged ticks of both the sampled species - H. longicornis and R. microplus. Partial nucleotide sequences of the S segment of the viral genome were acquired and a simple phylogenetic analysis was performed.

The manuscript is in general well written and comprehensive. Nevertheless, I have several major comments which are, from my point of view, necessary to be addressed.

Major comments:

  1. Please add information so far known on the ecology of the virus. You state, that this information is available and it may be important when interpreting your results. In fact, is the virus really vector-borne? Does it infect other hosts except the ticks?

  1. Since there is currently no indication that DTV infects humans (or animals?) and is able to cause disease, I would strongly recommend avoiding any speculations on its significance for public health, avoid using the terms epidemiological. Or support your statements with adequate evidence of epidemiological relevance.

  1. Additional information needed in the methods section: Please, be more specific, how was the tick species identification confirmed: which genes precisely, were they sequenced or species-specific primers? In any case, please, include the sequences of primers. Were the isolated RNAs used for this confirmation? Was the confirmation performer for all the samples or only for a subset? How many ticks from which location, how many samples by dragging and collected on the host?

  1. Please add the information on tick developmental stage and DTV prevalence according to tick developmental stage. The prevalence may differ dramatically for some microorganisms.

  1. Please, provide a more complex phylogenetic tree including other members of the genus. This tree is not very informative – all it basically says is, that the sequences acquired in this study are more closely related to previously published DTV sequences than to a single arbitrarily picked Yongjia virus sequence. Furthermore, concerning the fact, that this I first report of DTV in R. microplus it is particularly important to show, that the detected virus is really DTV and not for instance Lihan uukuvirus which is known to be found in R. microplus.

  1. The discussion section can be further improved. Items to be addressed:

  1. DTV in R. microplus – why wasn´t it detected before? Are there any other closely related viruses found in this ticks species? May they be misidentified?
  2. Unsuccessful isolation attempts – why BHK cells? Any indication, they should be permissive to infection? What about trying tick cell lines?

Abstract:

Line 14:  which becomes a potential threat to public health and safety => and becomes a potential threat to public health and safety? More importantly, this statement is highly speculative as there are no human clinical cases recorded so far. Please, delete or at least add the information, that no cases were reported.

Line 19 and elsewhere: species names lower case R. Microplus => R. microplus

Introduction:

Line 28: Tick-borne viruses (TBVs) is a large group of viruses => Tick-borne viruses (TBVs) are a large group of viruses

Line29: the sentence is hard to understand, please rephrase; e.g.: Currently, the TBVs known to cause human disease belong mainly to five families, such as Nairoviridae, Phenuiviridae, Flaviviridae, Orthomyxoviridae and Reoviridae.

Line 36: in varying degrees of disease => in disease of varying severity?

Line 41, 42: citations are missing + information on the probable/possible zoonotic reservoir of the virus; this might be also important considering the virus isolation attempts – did you consider using tick cell lines or cell lines of the probable vertebrate host?

Line 44: sequence => nucleotide sequence

Line 46: again, highly speculative, there are numerous arthropod associated viruses, that are not transmitted to humans and/or do not cause disease; I would suggest to delete this section.

Line 51: whchi => which

Line 54: existence => occurrence

Line 56: So far, there is no indication that „The presence of this virus may pose a threat to local public health.“

Results

Please add information on the developmental stage of the sampled ticks.

Line 65 store => stored; Nevertheless, I suggest deleting this information as it belongs to the Methods section

Table 1. bold either for the whole first two rows? R. Microplus, H. Longicornis  => R. microplus, H. longicornis; Grassland => dragging?

Line 76: were => was

Line 87: sequence => sequences

Line 94: isolates => sequences

Figure 2 (=> in fact, it should be Fig 1): Please, provide a more complex phylogenetic tree including other members of the genus. Concerning the fact, that this I first report of DTV in R. microplus  it is particularly important to show, that the detected virus is really DTV and not, for instance, Lihan uukuvirus which is known to be found in R. microplus. This tree is not very informative. Also, please describe the method in more detail: e.g. substitution model used, GenBank accession numbers for downloaded sequences, how was the statistical support for the nodes calculated, what is the length of branches corresponding to, etc.

Discussion:

Line 107: TBVDs – abbreviation used for the first time, please use the full form

Methods

Line 148: Auguest => August

Line 150: Figure 1 => Figure 2

Line 155, 156: Please, be more specific, how was the tick species identification confirmed: which genes precisely, were they sequenced or species-specific primers? In any case, please, include the sequences of primers. Were the isolated RNAs used for this confirmation? Was the confirmation performer for all the samples or only for a subset? How many ticks from which location, how many samples by dragging and collected on the host?

Line 158: pooled individually => processed individually

Line 162: specimen was => specimens were

Line 162. How were the ticks homogenized?

Line 163: Then, the clarified supernatant of tick homogenates were prepared for RNA extraction using TIANamp RNA extraction kit (Tiangen, China) following the manufacturer’s instructions. => Then, the clarified supernatants of tick homogenates were used for RNA extraction using TIANamp RNA 164 extraction kit (Tiangen, China) following the manufacturer’s instructions.

Line 168: in the RNA extraction from individual pools., please delete, it is an obsolete information

Line :  targeted => targeting

Line 169: I would suggest modifying as follows or prepare an additional table with sequences of primers and probes (including primers for tick species identity confirmation).

„Primer sets and a probe targeting the S segment of DTV used in this study: QRT-PCR primer sets DTV-F and DTV-R (TGC TCC TCT CCG CAC ACC T and TGG CAA GTA GAG GAA ACT GGT GA, respectively)….“

Line 183: detected => tested for the presence of DTV RNA

Conclusions

Line 196: Please, correct the numbering in the subtitle: 1. Conclusions => 4. Conclusions

Line 197: „epidemiological“ – I strongly suggest deleting this as there is currently no indication that the virus is able to infect people or cause disease in animals

Author Response

Major comments:

  1. Please add information so far known on the ecology of the virus. You state, that this information is available and it may be important when interpreting your results. In fact, is the virus really vector-borne? Does it infect other hosts except the ticks?

Response: Thank you for the comment. There are few researches on DTV which was first identified in Haemaphysalis longicornis by high-throughput sequencing in 2015 and no virus has been successfully isolated until now. We have tested the occurrence of DTV in about 1000 mosquitos (divided into 67 pools) collected during this study and no positive result was obtained (not mentioned in this manuscript).

  1. Since there is currently no indication that DTV infects humans (or animals?) and is able to cause disease, I would strongly recommend avoiding any speculations on its significance for public health, avoid using the terms epidemiological. Or support your statements with adequate evidence of epidemiological relevance.

Response: Thank you for the comment. We have revised the text accordingly.

  1. Additional information needed in the methods section: Please, be more specific, how was the tick species identification confirmed: which genes precisely, were they sequenced or species-specific primers? In any case, please, include the sequences of primers. Were the isolated RNAs used for this confirmation? Was the confirmation performer for all the samples or only for a subset? How many ticks from which location, how many samples by dragging and collected on the host?

Response: Thank you for the comment. We have revised the description of species identification in the Methods section. It now reads: " we preliminarily determined the species of ticks according to morphology, and the result was verified by the detection of the tick conserved gene: COI. PCR primer sets used in this study: LCO1490- 5'-ggtcaacaaatcataaagatattgg-3'and HCO2198- 5'-taaacttcagggtgaccaaaaaatca-3' ”. Because there are obvious differences between Haemaphysalis longicornis and Haemaphysalis, we first classify all ticks morphologically, and then select representative subsets for confirmation. In addition, the information on tick collection is shown in Table 1.

  1. Please add the information on tick developmental stage and DTV prevalence according to tick developmental stage. The prevalence may differ dramatically for some microorganisms.

Response: Thank you for the comment. We have revised the description of stage of collected ticks in Results section. They now read: " In total, 116 ticks of the species H. longicornis (95, 81.9%) and R. microplus (21, 18.1%) were collected from cattle hosts. All ticks collected from animals were engorged adults. Meanwhile, 273 ticks were collected by dragging a cloth through vegetation, including H. longicornis (224, 82.1%, 224 adults, 0 nymphs, and 0 larvae) and R. microplus (49, 17.9%, 49 adults, 0 nymphs, and 0 larvae)”. While due to the reason that the ticks were collected during July to August, all samples were adult.

  1. Please, provide a more complex phylogenetic tree including other members of the genus. This tree is not very informative – all it basically says is, that the sequences acquired in this study are more closely related to previously published DTV sequences than to a single arbitrarily picked Yongjia virus sequence. Furthermore, concerning the fact, that this I first report of DTV in R. microplus it is particularly important to show, that the detected virus is really DTV and not for instance Lihan uukuvirus which is known to be found in R. microplus.

Response: Thank you for the comment. We have added another phylogenetic tree including other Uukuvirus virus members as shown in Figure 2.

  1. The discussion section can be further improved. Items to be addressed:

  1. DTV in R. microplus – why wasn´t it detected before? Are there any other closely related viruses found in this ticks species? May they be misidentified?
  2. Unsuccessful isolation attempts – why BHK cells? Any indication, they should be permissive to infection? What about trying tick cell lines?

Response: Thank you for the comment. There are few researches on DTV which was first identified in Haemaphysalis longicornis by high-throughput sequencing in 2015 and no virus has been successfully isolated until now. The sequence of successfully amplified partial S segment (548 bp) have been blasted in NCBI website and confirmed as DTV, not other tick-borne virus. In fact, Vero cells and C6/36 cells were also used in this study (not mentioned in the manuscript), while no virus was still successfully isolated and there were no tick cell lines or cell lines of the probable vertebrate host available for us at that time.

Abstract:

Line 14:  which becomes a potential threat to public health and safety => and becomes a potential threat to public health and safety? More importantly, this statement is highly speculative as there are no human clinical cases recorded so far. Please, delete or at least add the information, that no cases were reported.

Response: Thank you for the comment. We have deleted this sentence.

Line 19 and elsewhere: species names lower case R. Microplus => R. microplus

Response: Thank you for the comment. We have revised the text accordingly.

Introduction:

Line 28: Tick-borne viruses (TBVs) is a large group of viruses => Tick-borne viruses (TBVs) are a large group of viruses

Response: Thank you for the comment. We have corrected this clerical mistake.

Line29: the sentence is hard to understand, please rephrase; e.g.: Currently, the TBVs known to cause human disease belong mainly to five families, such as Nairoviridae, Phenuiviridae, Flaviviridae, Orthomyxoviridae and Reoviridae.

Response: Thank you for the comment. We have revised the sentence accordingly.

Line 36: in varying degrees of disease => in disease of varying severity?

Response: Thank you for the comment. We have revised the text accordingly.

Line 41, 42: citations are missing + information on the probable/possible zoonotic reservoir of the virus; this might be also important considering the virus isolation attempts – did you consider using tick cell lines or cell lines of the probable vertebrate host?

Response: Thank you for the comment. There are few references on DTV and some have been cited in other text of this manuscript. In fact, Vero cells and C6/36 cells were also used in this study (not mentioned in the manuscript), while no virus was still successfully isolated and there were no tick cell lines or cell lines of the probable vertebrate host available for us at that time.

Line 44: sequence => nucleotide sequence

Response: Thank you for the comment. We have revised the text accordingly.

Line 46: again, highly speculative, there are numerous arthropod associated viruses, that are not transmitted to humans and/or do not cause disease; I would suggest to delete this section.

Response: Thank you for the comment. We have deleted this sentence accordingly.

Line 51: whchi => which

Response: Thank you for the comment. We have corrected this clerical mistake.

Line 54: existence => occurrence

Response: Thank you for the comment. We have revised the text accordingly.

Line 56: So far, there is no indication that „The presence of this virus may pose a threat to local public health.“

Response: Thank you for the comment. We have deleted this sentence accordingly.

Results

Please add information on the developmental stage of the sampled ticks.

Response: Thank you for the comment. We have added relevant information. Now it read: “In total, 116 ticks of the species H. longicornis (95, 81.9%) and R. microplus (21, 18.1%) were collected from cattle hosts. All ticks collected from animals were engorged adults. Meanwhile, 273 ticks were collected by dragging a cloth through vegetation, including H. longicornis (224, 82.1%, 224 adults, 0 nymphs, and 0 larvae) and R. microplus (49, 17.9%, 49 adults, 0 nymphs, and 0 larvae)”.

Line 65 store => stored; Nevertheless, I suggest deleting this information as it belongs to the Methods section

Response: Thank you for the comment. We have corrected this clerical mistake and revised the corresponding sentences.

Table 1. bold either for the whole first two rows? R. Microplus, H. Longicornis  => R. microplus, H. longicornis; Grassland => dragging?

Response: Thank you for the comment. We have revised the text accordingly.

Line 76: were => was

Response: Thank you for the comment. We have corrected this clerical mistake.

Line 87: sequence => sequences

Response: Thank you for the comment. We have corrected this clerical mistake.

Line 94: isolates => sequences

Response: Thank you for the comment. We have revised the texts accordingly.

Figure 2 (=> in fact, it should be Fig 1): Please, provide a more complex phylogenetic tree including other members of the genus. Concerning the fact, that this I first report of DTV in R. microplus  it is particularly important to show, that the detected virus is really DTV and not, for instance, Lihan uukuvirus which is known to be found in R. microplus. This tree is not very informative. Also, please describe the method in more detail: e.g. substitution model used, GenBank accession numbers for downloaded sequences, how was the statistical support for the nodes calculated, what is the length of branches corresponding to, etc.

Response: Thank you for the comment. We have redrawn the phylogenetic tree including other members of Uukuvirus genus. The sequence of successfully amplified partial S segment (548 bp) have been blasted in NCBI website and confirmed as DTV, not Lihan uukuvirus.We also have revised the description of the method and now it reads: “MEGA5.1 software was utilized for phylogenetic analysis by constructing the neighbor-joining evolutionary tree with 1,000 replicates, compared with previous published viral sequences of DTV strains and other Uukuvirus genus strains, such as Rukutama virus, Kabuto mountain virus, Uukuniemi virus, Silverwater virus and so on. The relevant GenBank accession numbers has been marked in Figure 2.”.

Discussion:

Line 107: TBVDs – abbreviation used for the first time, please use the full form

Response: Thank you for the comment. We have revised the text accordingly.

Methods

Line 148: Auguest => August

Response: Thank you for the comment. We have corrected this clerical mistake.

Line 150: Figure 1 => Figure 2

Response: Thank you for the comment. We have corrected this clerical mistake.

Line 155, 156: Please, be more specific, how was the tick species identification confirmed: which genes precisely, were they sequenced or species-specific primers? In any case, please, include the sequences of primers. Were the isolated RNAs used for this confirmation? Was the confirmation performer for all the samples or only for a subset? How many ticks from which location, how many samples by dragging and collected on the host?

Response: Thank you for the comment. The tick species were identified by PCR with species-specific primers using the isolated DNAs. We have added the information in the Methods section and now it reads: “we preliminarily determined the species of ticks according to morphology, and the result was verified by the detection of the tick conserved gene: COI. PCR primer sets used in this study: LCO1490- 5'-ggtcaacaaatcataaagatattgg-3'and HCO2198- 5'-taaacttcagggtgaccaaaaaatca-3' ”.The number of ticks collected has been described in the Results section.

Line 158: pooled individually => processed individually

Response: Thank you for the comment. We have revised the text accordingly.

Line 162: specimen was => specimens were

Response: Thank you for the comment. We have revised the text accordingly.

Line 162. How were the ticks homogenized?

Response: Thank you for the comment. We have added the description how the ticks were homogenized. Now it reads: “Individual or pooled tick specimens were surface-sterilized with sequential washes with Dulbecco's modified eagle's medium (DMEM) containing antibiotics and then homogenized in 500 μL chilled DMEM using a tissue homogenizer (Qiagen, Germany).”.

Line 163: Then, the clarified supernatant of tick homogenates were prepared for RNA extraction using TIANamp RNA extraction kit (Tiangen, China) following the manufacturer’s instructions. => Then, the clarified supernatants of tick homogenates were used for RNA extraction using TIANamp RNA 164 extraction kit (Tiangen, China) following the manufacturer’s instructions.

Response: Thank you for the comment. We have revised the corresponding sentence.

Line 168: in the RNA extraction from individual pools., please delete, it is an obsolete information

Response: Thank you for the comment. We have deleted the corresponding text.

Line169:targeted => targeting

Response: Thank you for the comment. We have corrected this clerical mistake.

Line 169: I would suggest modifying as follows or prepare an additional table with sequences of primers and probes (including primers for tick species identity confirmation).

 „Primer sets and probe targeting the S segment of DTV used in this study: QRT-PCR primer sets DTV-F and DTV-R (TGC TCC TCT CCG CAC ACC T and TGG CAA GTA GAG GAA ACT GGT GA, respectively)….“

Response: Thank you for the comment. We have revised the description and added the primers and probes for tick species identification.

Line 183: detected => tested for the presence of DTV RNA

Response: Thank you for the comment. We have revised the text accordingly.

Conclusions

Line 196: Please, correct the numbering in the subtitle: 1. Conclusions => 4. Conclusions

Response: Thank you for the comment. We have corrected this clerical mistake.

Line 197: „epidemiological“ – I strongly suggest deleting this as there is currently no indication that the virus is able to infect people or cause disease in animals

Response: Thank you for the comment. We have revised this sentence accordingly.

Reviewer 2 Report

The work is interesting and original, and needs the following adjustments:
In the abstract R Microplus and H. Longicornis R. microplus and H. longicornis should be corrected.
In the Introduction page 1 line 39-40 (Dabieshan tick virus ....... from Hubei Province in 2015.), it is repeated in the discussion page 3 line 107-109.
It should be specified whether the Haemaphysalis longicornis ticks (which can also transmit Rickettsia rickettsii the Agent of Rocky Mountain Spotted Fever) and Rhipicephalus (Boophilus) microplus (which can also transmit Babesia bovis and B. bigemina the Agents of bovine babesiosis) also attack humans .
Page 2 line 51 whchi is which
Tick collection line 59-70 are Materials and Methods and must be assembled with Methods page 4-6 line 146-195.
Materials and Methods go before the Results, which are page 2-3 line 71-100,
Discussion and Conclusions after the Results.

Author Response

In the abstract R Microplus and H. Longicornis R. microplus and H. longicornis should be corrected.

Response: Thank you for the comment. We have revised the text accordingly.

In the Introduction page 1 line 39-40 (Dabieshan tick virus ....... from Hubei Province in 2015.), it is repeated in the discussion page 3 line 107-109.

Response: Thank you for the comment. We have deleted the corresponding sentence in the discussion section.

It should be specified whether the Haemaphysalis longicornis ticks (which can also transmit Rickettsia rickettsii the Agent of Rocky Mountain Spotted Fever) and Rhipicephalus (Boophilus) microplus (which can also transmit Babesiabovis and B. bigemina the Agents of bovine babesiosis) also attack humans.

Response:Thank you for the comment.We have added thei nformation about whether both kinds of ticks bite human in the discussion section.

Page 2 line 51 whchi is which

Response: Thank you for the comment. We have corrected this clerical mistake.

Tick collection line 59-70 are Materials and Methods and must be assembled with Methods page 4-6 line 146-195.

Response:Thank you for the comment. We have revised the corresponding sentences.

Materials and Methods go before the Results, which are page 2-3 line 71-100,Discussion and Conclusions after the Results.

Response: Thank you for the comment.The research manuscript sections of this journal are as follows: Introduction, Results, Discussion, Materials and Methods, Conclusions.

Reviewer 3 Report

Comments:

  • The warning about "a potential threat" of DTV for humans and animals is repeated many times although there is no data concerning a real threat. Mentioning one time in the Abstract and one time in Introduction would be enough.
  • When you mention a tick species for the first time, you must give the full name (Haemaphysalis longicornis, Rhipicephalus (Boophilus) microplus). The species should be write with a lowercase letter - longicornis, microplus (see e.g. lines 19, 199, Table 1).
  • Tick names should be given in Key words.
  • Lines 34-35 - The sentence is rather vague. It is possible to omit it.
  • L 48 - "virological (what?) of the virus - absurdity.
  • 2.1. Tick collection - what about the sex and stage of the ticks collected (on line 157 you mention the stage)?
  • L 86 - the Uukuniemi virus is not shown in the Figure.
  • Figure 2 - where is Fig. 1?
  • L 104 - ALSV was discovered not only in China, but also in France, Finland, many regions of Russia (Kuivanen et al., 2019; Kholodilov et al., 2021).
  • L 111 - "low population density" - human?
  • L 119 - "high prevalence of DTV in engorged ticks" - it would be appropriate to draw a parallel with the virus of tick-borne encephalitis.
  • L 128 - "potentially pathogenic" - you must prove it; for the time being there is no data.
  • L 129-139 and L 140-145 contradict each other. It is impossible to do any conclusions with so strong limitations of your study.
  • L 153-154 - Collecting ticks by flagging or dragging are so routine techniques used for a century that it is not necessary to cite [16] any authors.
  • There are many repetitions in the text. 

Author Response

The warning about "a potential threat" of DTV for humans and animals is repeated many times although there is no data concerning a real threat. Mentioning one time in the Abstract and one time in Introduction would be enough.

Response: Thank you for the comment. We have revised the text accordingly.

When you mention a tick species for the first time, you must give the full name (Haemaphysalislongicornis, Rhipicephalus (Boophilus) microplus). The species should be write with a lowercase letter - longicornis, microplus (see e.g. lines 19, 199, Table 1).

Response: Thank you for the comment. We have revised the text accordingly.

Tick names should be given in Key words. 

Response: Thank you for the comment. We have revised the text accordingly.

Lines 34-35 - The sentence is rather vague. It is possible to omit it.

Response: Thank you for the comment. We have revised the sentence.Now it reads:“These TBVs are transmitted from their natural hosts to human beings through zoonotic infection sources, resulting in disease of varying severity or even death.”.

L 48 - "virological (what?) of the virus - absurdity.

Response: Thank you for the comment. We have revised the text accordingly.Now it reads:“…the entomological, virological and epidemiological features of this virus…”.

2.1. Tick collection - what about the sex and stage of the ticks collected (on line 157 you mention the stage)?

Response: Thank you for the comment. Thank you for the comment. We have revised the description of stage and sex of collected ticks in Results section. They now read: " In total, 116 ticks of the species H. longicornis (95, 81.9%) and R. microplus (21, 18.1%) were collected from cattle hosts. All ticks collected from animals were engorged adults. Meanwhile, 273 ticks were collected by dragging a cloth through vegetation, including H. longicornis (224, 82.1%, 224 adults, 0 nymphs, and 0 larvae) and R. microplus (49, 17.9%, 49 adults, 0 nymphs, and 0 larvae)”. While due to the reason that the ticks were collected during July to August, all samples were adult. Different with mosquitos, both female and male ticks suck blood, so we think there is no need to distinguish their gender.

L 86 - the Uukuniemi virus is not shown in the Figure.

Response: Thank you for the comment.We have added relevant information in Figure 2.

Figure 2 - where is Fig. 1?

Response: Thank you for the comment. We have corrected this clerical mistake in the manuscript.

L 104 - ALSV was discovered not only in China, but also in France, Finland, many regions of Russia (Kuivanen et al., 2019; Kholodilov et al., 2021).

Response: Thank you for the comment.This sentence described the locations where these viruses were identified for the first time.

L 111 - "low population density" - human?

Response: Thank you for the comment.We have revised thissentence. Now it reads: “…a remote area with low human population density…”.

L 119 - "high prevalence of DTV in engorged ticks" - it would be appropriate to draw a parallel with the virus of tick-borne encephalitis.

Response: Thank you for the comment. In this section, we intend to compare the prevalence of DTV carried by engorged ticks and unfed ticks and analyse the role of sucking animal blood in virus replication . Hence, other viruses including tick-borne encephalitis virus were not mentioned.

L 128 - "potentially pathogenic" - you must prove it; for the time being there is no data.

Response: Thank you for the comment.We have deleted this sentence.

L 129-139 and L 140-145 contradict each other. It is impossible to do any conclusions with so strong limitations of your study.

Response: Thank you for the comment.We have revised the text accordingly.

L 153-154 - Collecting ticks by flagging or dragging are so routine techniques used for a century that it is not necessary to cite [16] any authors.

Response: Thank you for the comment.We have deleted this reference.
